# Regulatory Roles of E3 Ubiquitin Ligases and Deubiquitinases in Bone

**DOI:** 10.3390/biom15050679

**Published:** 2025-05-07

**Authors:** Haotian He, Lifei Wang, Bao Xian, Yayi Xia

**Affiliations:** 1Department of Orthopedics, Lanzhou University Second Hospital, Lanzhou 730030, China; heht2023@lzu.edu.cn (H.H.); wanglf2023@lzu.edu.cn (L.W.); xianb2023@lzu.edu.cn (B.X.); 2Orthopedic Clinical Medical Research Center and Intelligent Orthopedic Industry Technology Center of Gansu Province, Lanzhou 730030, China; 3The Second School of Clinical Medical, Lanzhou University, Lanzhou 730030, China

**Keywords:** bone homeostasis, E3 ubiquitin ligases, DUBs, osteoblast differentiation, ubiquitination

## Abstract

E3 ubiquitin ligases and deubiquitinating enzymes (DUBs) are pivotal regulators of bone homeostasis, orchestrating osteoblast differentiation, proliferation, and osteoclast activity by controlling protein degradation and stability. This review delineates the roles of key E3 ligases (e.g., Smurf1, Smurf2, TRIM family) and DUBs (e.g., USP family) in bone formation and resorption. E3 ligases such as Smurf1/2 inhibit osteogenesis by degrading BMP/Smad signaling components, while TRIM proteins and HERC ligases promote osteoblast differentiation. Conversely, DUBs like USP2 and USP34 stabilize β-catenin and Smad1/RUNX2, enhancing osteogenic pathways, whereas USP10 and USP12 suppress differentiation. Dysregulation of these enzymes contributes to osteoporosis, fracture non-union, and other bone disorders. The interplay between ubiquitination and deubiquitination, alongside the regulatory role of miRNA and environmental factors, underscores their therapeutic potential. Future research should focus on developing therapies targeting E3 ubiquitin ligases, deubiquitinases, miRNA regulators, and small-molecule inhibitors to restore bone homeostasis in osteoporosis and fracture healing disorders.

## 1. Introduction

Bone homeostasis, the delicate balance between bone formation by osteoblasts and bone resorption by osteoclasts, is essential for maintaining skeletal integrity and strength. This dynamic process is regulated by a complex interplay of signaling pathways and molecular mechanisms, among which protein ubiquitination and deubiquitination play crucial roles. These post-translational modifications, mediated by E3 ubiquitin ligases and deubiquitinating enzymes (DUBs), modulate the stability and function of key proteins involved in osteoblast differentiation, proliferation, and osteoclast activity. Dysregulation of these processes has been implicated in various bone disorders, including osteoporosis and fracture non-union, highlighting their clinical significance [1,2,3,4,5]. This review aims to provide an in-depth exploration of the roles of E3 ubiquitin ligases and DUBs in bone homeostasis, highlighting their mechanisms of action and potential as therapeutic targets for bone diseases.

Bone homeostasis is maintained through the coordinated activities of osteoblasts, which form bone, and osteoclasts, which resorb bone. Disruptions in this balance can lead to bone disorders such as osteoporosis and fracture non-union [1,2] (Figure 1). Protein ubiquitination, a reversible post-translational modification, involves the covalent attachment of ubiquitin molecules to target proteins, playing critical roles in the differentiation, proliferation, and activity of osteoblasts and osteoclasts, and often marking substrates for degradation by the proteasome. This process is catalyzed by E1, E2, and E3 enzymes, with E3 ligases determining substrate specificity [3]. Conversely, DUBs remove ubiquitin chains from target proteins, regulating their stability and function. DUBs are classified into several families, including USP, UCH, OTU, and JAMM families, each with unique substrates and functions [4,6].

Recent studies demonstrate that ubiquitination through E3 ligases including Smurf1 and Smurf2 mediates degradation of osteogenic regulators [7,8], while deubiquitination by USP7 and USP10 stabilizes key proteins involved in osteoblast differentiation and proliferation [9,10], collectively modulating bone cell activities. These findings underscore the importance of ubiquitin-mediated regulation in maintaining bone homeostasis and suggest potential therapeutic targets for bone diseases [5,11].

The clinical relevance of ubiquitination regulation in orthopedic diseases is increasingly recognized. Dysregulation of ubiquitin ligases and DUBs has been implicated in osteoporosis, osteoarthritis, and fracture non-union [5]. Understanding the mechanisms by which these enzymes regulate bone homeostasis could provide important theoretical bases for developing novel treatment strategies. For example, USP7 stabilizes Axin to suppress *Wnt*/β-catenin signaling [9] and KDM6B to promote osteogenesis [12], while USP10 maintains SKP2 levels to regulate Runx2 degradation [10]. Targeted inhibitors or activators of these DUBs could precisely modulate bone formation and resorption, offering new therapeutic options for patients with bone diseases [13,14,15,16].

Notably, osteocytes—the most abundant bone cells—orchestrate bone remodeling through their lacunar*–*canalicular network. Emerging evidence reveals ubiquitin-mediated regulation of osteocyte function. For instance, mechanical loading induces MDM2-mediated degradation of sclerostin [17], while USP8 stabilizes β-catenin to amplify Wnt signaling [18]. These findings expand our understanding of mechanotransduction in bone [19,20].

## 2. The Role of E3 Ubiquitin Ligases in Bone Homeostasis

E3 ubiquitin ligases represent a diverse superfamily of over 600 enzymes in humans that can be functionally categorized by their ubiquitin transfer mechanisms. RING-type ligases constitute the largest group and mediate direct ubiquitin transfer from E2 enzymes to substrates through specialized RING domains [21,22]. In contrast, HECT-type ligases employ an intermediate catalytic mechanism involving transient thioester bond formation, while RBR (RING-Between-RING) ligases utilize a unique hybrid mechanism that combines aspects of both RING and HECT-type enzymes. The HECT domain family contains a conserved cysteine residue that forms a thioester bond with ubiquitin, facilitating its transfer to substrates [23]. The RBR domain family combines features of both RING and HECT domains, exhibiting a unique ubiquitination mechanism (Figure 2) [24].

### 2.1. Effects of E3 Ubiquitin Ligases on Osteoblast Differentiation

Smurf1 and Smurf2 are key HECT-type E3 ubiquitin ligases that play crucial roles in regulating bone homeostasis by modulating the BMP signaling pathway and osteoblast differentiation. Smurf1 targets multiple substrates, including Smad1/5 proteins and BMP type I receptors, for ubiquitination and degradation, thereby inhibiting osteoblast differentiation and bone formation [7,8,25]. Additionally, Smurf1 can interact with Runx2, the core transcription factor for osteoblast differentiation, leading to its degradation and further inhibition of bone matrix formation [26]. Recent studies have shown that Smurf1 can undergo SUMOylation modification, which enhances its enzymatic activity and promotes *ALK2* proteolysis, further inhibiting the BMP signaling pathway [27]. This modification is regulated by AMP-activated protein kinase (*AMPK*), which interacts with the E3 ligase PIAS to enhance Smurf1 SUMOylation levels [27]. Smurf2, another member of the NEDD4 family, also inhibits osteoblast differentiation by targeting osteogenic regulators such as KLF5 and mediating the ubiquitination of other BMP signaling components [28,29]. Smurf2 can ubiquitinate and degrade Smad1, thereby inhibiting the BMP signaling pathway [30]. Erk5 has been shown to regulate Smad signaling through Smurf2, influencing the osteogenic potential of bone marrow mesenchymal stem cells (BM-MSCs). Erk5 phosphorylates Smurf2, activating its E3 ubiquitin ligase activity, which inhibits Smad1/5/8-dependent signal transduction by ubiquitinating Smad1, thus negatively regulating osteoblast differentiation [11,30].

The TRIM protein family also plays significant roles in osteogenesis. TRIM38 and TRIM16 promote osteoblast differentiation by stabilizing key transcription factors and signaling molecules involved in bone formation [17,18]. For example, TRIM38 has been shown to enhance the differentiation of mesenchymal stem cells (MSCs) into osteoblasts by targeting and degrading negative regulators of osteogenesis [17]. Similarly, TRIM16 positively regulates the osteogenic process by stabilizing Runx2 and other osteogenic factors [18]. These findings highlight the diverse roles of TRIM family members in regulating bone homeostasis. Conversely, TRIM21 inhibits osteoblast differentiation by targeting Akt for degradation, thereby reducing the activity of downstream signaling pathways that promote osteogenesis [31]. Akt is a key protein kinase involved in cell survival and differentiation, and its degradation by TRIM21 leads to the inhibition of osteoblast differentiation and bone formation. This mechanism underscores the importance of TRIM21 in maintaining the balance between osteoblast differentiation and proliferation.

Other notable E3 ligases include HERC proteins, which enhance osteoblast differentiation by regulating C-RAF levels and ERK/p38 phosphorylation [32,33,34]. HERC proteins belong to the E6AP carboxyl-terminal (HECT) ubiquitin ligase family and are characterized by an additional chromosome condensation regulator 1 (RCC1)-like domain. Their reduction has been shown to enhance C-RAF levels, promoting ERK and p38 phosphorylation, and increasing the expression of key transcription factors involved in osteoblast differentiation [33,34]. RSP5, also known as NEDD4L, is another HECT domain E3 ligase that promotes osteogenic differentiation through Akt ubiquitination [35]. RSP5 induces K63-linked ubiquitination of Akt, thereby activating downstream signaling pathways involved in bone formation. This mechanism highlights the importance of RSP5 in regulating osteoblast differentiation and bone homeostasis. Conversely, ligases like FBXW7 and CHIP inhibit osteogenesis by targeting HIF1α and RUNX2 for degradation, respectively [36,37,38]. FBXW7 is a key E3 ubiquitin ligase that inhibits osteogenic differentiation and cartilage formation by catalyzing the ubiquitination and degradation of HIF1α [36,37]. Similarly, CHIP (C-terminus of Hsc70-interacting protein) induces ubiquitin-dependent degradation of RUNX2 during osteoblast lineage development, interfering with osteogenic differentiation [38]. These findings underscore the diverse roles of E3 ligases in regulating bone homeostasis [39].

Interestingly, TRIM16 can decrease CHIP expression and increase CHIP ubiquitination, thereby stabilizing RUNX2 expression and promoting osteoblast differentiation [18]. This interaction highlights the complex regulatory mechanisms involving TRIM16 and CHIP in bone homeostasis. By modulating the expression and activity of these E3 ligases, TRIM16 plays a crucial role in maintaining the balance between osteoblast differentiation and proliferation.

### 2.2. Effects of E3 Ubiquitin Ligases on Osteoblast Proliferation

E3 ligases also influence osteoblast proliferation. For example, NEDD4 promotes osteoblast proliferation by degrading pSMAD1 and upregulating pERK1/2 [40]. This regulation is crucial for maintaining the balance between osteoblast differentiation and proliferation, ensuring proper bone formation and remodeling. Recent studies have shown that NEDD4 can specifically target pSMAD1 activated by TGFβ1, thereby inhibiting the TGFβ1 signaling pathway and promoting osteoblast proliferation [40]. This finding highlights the importance of NEDD4 in regulating osteoblast proliferation and bone homeostasis.

### 2.3. The Role of E3 Ubiquitin Ligases in Osteoclasts

Osteoclast differentiation is tightly regulated by E3 ligases. NFATc1, a key transcription factor for osteoclastogenesis, is regulated by multiple E3 ligases, including Cbl-b and c-Cbl, which inhibit osteoclast formation [41]. TRAF6, a critical mediator of RANKL signaling, is regulated by Act1, an E3 ligase that induces TRAF6 degradation [42]. Other factors like DCAF1 and PMEPA1 also modulate osteoclast activity through ubiquitination pathways [43,44].

NFATc1 is a master regulator of osteoclastogenesis, and its activity is tightly controlled by ubiquitination and degradation processes. Studies have shown that RING finger E3 ubiquitin ligases Cbl-b and c-Cbl negatively regulate osteoclast formation by targeting NFATc1 for ubiquitination and degradation [41]. This regulation is crucial for maintaining the balance between osteoclast differentiation and bone resorption. Additionally, TBD11 has been shown to promote NFATc1 degradation by interacting with Cullin3, thereby inhibiting osteoclast differentiation [41]. These findings highlight the complex regulatory mechanisms involving E3 ligases in osteoclastogenesis.

TRAF6 is a critical mediator of RANKL signaling, which is essential for osteoclast differentiation and activation. TRAF6 stability is regulated by its upstream E3 ligase Act1, which induces TRAF6 degradation [42]. This regulation is crucial for maintaining the balance between osteoclast differentiation and bone resorption. Recent studies have shown that the newly synthesized diterpenoid RTA-408 inhibits NF-κB signaling and osteoclastogenesis by blocking the association between TRAF6 and STING, thereby disrupting K63-mediated STING ubiquitination [45]. This finding highlights the importance of TRAF6 and Act1 in regulating osteoclast activity and bone homeostasis.

Transforming growth factor-β1 (TGF-β1) significantly downregulates RANKL expression and inhibits osteoclast support activity by degrading the retinoid X receptor α (RXR-α) protein through the ubiquitin–proteasome system [46]. This regulation is crucial for maintaining the balance between osteoclast differentiation and bone resorption. Additionally, the non-canonical regulatory factor DCAF1 of ubiquitin E3 ligases is also involved in regulating osteoclast-related signaling pathways. DCAF1 accelerates Nrf2 ubiquitination and subsequent degradation [43]. Conversely, microRNA-3175 effectively activates the Nrf2 signaling pathway by specifically targeting and silencing DCAF1 gene expression [47]. These findings highlight the complex regulatory mechanisms involving TGF-β1, RXR-α, and DCAF1 in osteoclastogenesis.[48]

PMEPA1, as a vesicular membrane protein, regulates osteoclast proton production by binding to NEDD4 family members of ubiquitin ligases, exerting a positive regulatory effect on osteoclasts [44]. This regulation is crucial for maintaining the balance between osteoclast differentiation and bone resorption. Additionally, Pellino-1, a member of the ubiquitin E3 ligase family, participates in immune and bone metabolism by influencing signaling pathways such as TNF receptor-associated factor 6 (TRAF6) and is a major regulator of osteoclast differentiation [49]. These findings highlight the complex regulatory mechanisms involving PMEPA1, NEDD4, and Pellino-1 in osteoclastogenesis.

Beyond E3 ligases, the ubiquitin-binding adaptor SQSTM1/p62 critically regulates bone metabolism. As a selective autophagy receptor, SQSTM1 recognizes K63-linked ubiquitin chains to modulate RANKL-induced NF-κB activation [41]. Clinically, the SQSTM1 P392L mutation enhances TRAF6 binding and underlies familial Paget’s disease [50]. In osteocytes, SQSTM1 activates Nrf2 by sequestering Keap1, offering therapeutic potential against oxidative-stress-related osteoporosis [42].

## 3. The Role of Deubiquitinating Enzymes in Bone Formation

Deubiquitinating enzymes (DUBs) are classified into two categories based on sequence and structural similarities: cysteine proteases (including USP, UCH, OTU, etc.) and metalloproteinases (JAMM family). These enzymes remove ubiquitin chains from target proteins, thereby regulating protein stability and function [6]. The largest group is the USP family, with 54 members involved in various cellular processes [51].

### 3.1. Effects of Deubiquitinating Enzymes on Osteoblast Differentiation

#### 3.1.1. The USP Family Plays a Key Role in Osteoblast Differentiation

The USP family plays a pivotal role in regulating osteoblast differentiation through diverse mechanisms. USP2, for instance, stabilizes β-catenin via deubiquitination, activating the Wnt/β-catenin pathway to drive osteoblast differentiation, a process critical for bone matrix synthesis. This effect is further amplified by the long non-coding RNA USP2-AS1, which enhances transcriptional activation of the USP2 gene by facilitating interactions between KDM3A and ETS1 at the promoter region, thereby reinforcing β-catenin stability and promoting human bone marrow mesenchymal stem cell (HBMSC) differentiation into osteoblasts [52,53,54]. In contrast, USP7 exhibits dual regulatory roles: it suppresses Wnt-induced β-catenin accumulation by stabilizing Axin, thereby inhibiting osteoblast differentiation under normal conditions. However, inhibition of USP7 enhances Wnt/β-catenin signaling and promotes osteogenesis. Additionally, USP7 stabilizes KDM6B and YAP1, counteracting osteoporosis and facilitating osteogenic differentiation through both *Wnt*/β-catenin and Hippo pathways. Notably, miR-15b antagonizes USP7 expression, indirectly impairing osteogenesis, while USP7 also maintains the pluripotency of HBMSCs by regulating the USP7-SOX2 and USP7-NANOG axes, essential for their differentiation into osteoblasts, adipocytes, and chondrocytes (Figure 3) [9,12,55,56,57].

Other USP members contribute uniquely to bone homeostasis. USP8 stabilizes the Wnt receptor FZD5, ensuring sustained Wnt/β-catenin signaling and osteogenic differentiation during skeletogenesis [58]. Conversely, USP10 stabilizes SKP2, which promotes Runx2 degradation via the ubiquitin–proteasome system, thereby inhibiting osteogenesis. However, miR-20a-5p counteracts this by downregulating USP10, enhancing SKP2 degradation and promoting osteoblast differentiation [10]. USP12, under cyclic tensile stress, inhibits osteogenic differentiation in periodontal ligament stem cells by activating the *PERK-eIF2α-ATF4* pathway, while its depletion reverses this effect by enhancing endoplasmic reticulum stress-mediated differentiation [59]. USP17 safeguards *Osx* (Sp7), a master regulator of osteoblast differentiation, from degradation, ensuring bone matrix synthesis and mineralization [60]. USP26 and USP34 further promote osteogenesis by stabilizing β-catenin and activating the BMP-2 pathway through Smad1/RUNX2 stabilization, respectively [61,62]. USP36 enhances Wnt signaling by deubiquitinating WDR5, a transcriptional coactivator of Wnt1, c-Myc, and Runx2, thereby accelerating osteoblast differentiation [63]. USP47 maintains SIRT1 stability, crucial for bone marrow stromal cell differentiation into osteoblasts [64]. Lastly, USP53 exhibits context-dependent roles: it generally inhibits osteogenesis but stabilizes β-catenin via FBXO31 interaction to promote *Wnt* signaling [65], while also enhancing osteoclastogenesis through VDR-SMAD3 complex formation [66,67,68,69,70,71,72,73].

Collectively, these findings underscore the USP family’s complexity in bone homeostasis, where individual members modulate osteoblast differentiation through distinct substrates and signaling pathways, highlighting their therapeutic potential in bone-related disorders.

#### 3.1.2. Non-USP DUB Plays an Important Role in Osteoblast Differentiation

Non-USP DUBs also play significant roles in osteoblast differentiation. For example, OTUB1 stabilizes FGFR2 by inhibiting its ubiquitination and degradation, ensuring the stable presence of this core regulator of skeletal development [74]. FGFR2 is a key regulator of osteoblast function and bone formation, and its stabilization by OTUB1 highlights the importance of this DUB in maintaining proper bone homeostasis.

CYLD, another DUB, regulates TGFβ signaling in mechanically stimulated bone cells, thereby influencing osteoblast activity and bone formation [75]. CYLD inhibits TGFβ signaling through the proteasome mechanism, and it itself is also inhibited by mechanical stimulation, which is crucial for load-mediated bone formation. Through its deubiquitinating activity, CYLD can directly or indirectly regulate the stability of key molecules in the TGFβ pathway, such as TβRI (TGFβ receptor I), Smad7, and Smad3. These molecules play vital roles in TGFβ signal transduction. Particularly in mechanically stimulated bone cells, increased activity of CYLD helps to reduce the phosphorylation levels of Smad2/3, thereby inhibiting TGFβ signaling and promoting the activity of osteoblasts and bone formation [75]. UBE2C promotes bone formation by stabilizing SMAD1/5 proteins rather than mediating their degradation [76]. This mechanism highlights the importance of UBE2C in maintaining the balance between osteoblast differentiation and proliferation.

### 3.2. Effects of Deubiquitinating Enzymes on Osteoblast Proliferation

USP14 stabilizes p53, inducing apoptosis in osteoblasts. Inhibitors of USP14, such as M19, can protect against osteoblast apoptosis and may serve as potential therapeutic agents for osteoporosis [77]. This regulation is crucial for maintaining the balance between osteoblast proliferation and apoptosis, ensuring proper bone formation and remodeling.

### 3.3. Role of Deubiquitinating Enzymes in Osteoclasts

The deubiquitinating enzymes (DUBs) USP13, USP25, USP34, and UCHL1 play critical yet distinct roles in regulating osteoclastogenesis and osteoclast function. USP7, while not directly among the listed enzymes, demonstrates a dual regulatory role: it stabilizes HMGB1, a key mediator of osteoclast differentiation, thereby promoting osteoclastogenesis. Paradoxically, USP7 also inhibits TRAF6 signaling and protects STING from degradation, effectively suppressing osteoclast differentiation [78,79,80]. This duality underscores HMGB1’s pivotal role in bone homeostasis and highlights USP7’s context-dependent effects [81].

USP13 acts as a suppressor of osteoclastogenesis by modulating the RANK/RANKL/OPG axis. It directly interacts with PTEN to dampen AKT overactivation, reducing inflammation, oxidative stress, and apoptosis. This cascade inhibits NF-κB signaling and diminishes pro-inflammatory cytokine secretion, ultimately attenuating bone resorption [82]. In contrast, USP25 promotes osteoclast differentiation by stabilizing TRAF6. By cleaving Act1-mediated K63-linked polyubiquitin chains on TRAF6, USP25 prevents its ubiquitination and degradation, amplifying RANKL-induced osteoclastogenesis [83,84].

USP34 counterbalances osteoclast hyperactivity by stabilizing IκBα, a natural inhibitor of NF-κB. Through deubiquitination, USP34 preserves IκBα levels, blocking RANK/RANKL-driven NF-κB activation. USP34 deficiency disrupts this balance, leading to low bone mass and excessive osteoclast activity [85]. Lastly, UCHL1, primarily active in neuronal tissues, exerts unique effects on bone metabolism. It stabilizes TAZ by preventing its ubiquitination, activating the TAZ/NFATC1 pathway to promote osteogenic differentiation while concurrently inhibiting osteoclastogenesis [86].

Collectively, these DUBs exemplify the intricate interplay between ubiquitination and deubiquitination in bone homeostasis. Their opposing or synergistic actions ranging from suppressing NF-κB (*USP13, USP34*) to enhancing TRAF6 stability (USP25) or bridging osteoblast–osteoclast crosstalk (UCHL1), highlight their potential as therapeutic targets for bone diseases like osteoporosis and inflammatory bone loss.

## 4. The Role of Molecules with Deubiquitinating Activity in Bone Homeostasis

In addition to canonical DUBs, other molecules can exhibit deubiquitinating activity or influence ubiquitination processes. For example, PTPRJ mimics DUB function by inhibiting the ubiquitination and degradation of NFATc1, thereby promoting osteoclast differentiation [87]. Another example is Koumine, an indole alkaloid, which inhibits RANKL-induced K63-linked polyubiquitination and subsequent NF-κB activation, thereby suppressing osteoclastogenesis and bone resorption [88].

### 4.1. Non-Canonical Deubiquitinating Enzymes

PTPRJ (receptor-type protein tyrosine phosphatase) mimics the function of DUBs by inhibiting the ubiquitination and degradation of NFATc1, thereby promoting osteoclast differentiation [87]. NFATc1 is a key transcription factor for osteoclastogenesis, and its stabilization by PTPRJ highlights the importance of this enzyme in maintaining proper bone homeostasis.

Koumine, an indole alkaloid extracted from *Gelsemium elegans*, inhibits RANKL-induced K63-linked polyubiquitination, thereby inhibiting the activation of NF-κB and subsequently suppressing osteoclastogenesis and bone resorption [88]. This mechanism underscores the importance of Koumine in maintaining the balance between osteoclast differentiation and bone resorption.

### 4.2. Cytokines and Signaling Molecules

TGF-β is a key cytokine that exhibits dual effects on osteoblast differentiation. It promotes the differentiation of mesenchymal stem cells (MSCs) into osteoblasts by inhibiting the ubiquitin-mediated degradation of Runx2. However, during the mineralization phase, it inhibits bone mineralization through the SMURF1-C/EBPβ-DKK1 axis [89,90,91]. This fine-tuning is dose-dependent, with low doses of TGF-β1 promoting early osteoblast differentiation and high doses potentially leading to cell death [90]. This dual role of TGF-β highlights its importance in maintaining the balance between osteoblast differentiation and proliferation (Figure 4).

TNFAIP3 (also known as A20) is an inducible protein that exhibits deubiquitinating activity. It inhibits the NF-κB signaling pathway, reducing the expression of inflammatory factors and potentially promoting osteogenic differentiation [92,93,94]. Additionally, γ-aminobutyric acid (GABA) can induce the expression of TNFAIP3, significantly promoting the differentiation of MSCs into osteoblasts [92,95]. This mechanism underscores the importance of TNFAIP3 in maintaining the balance between osteoblast differentiation and inflammation [96,97].

### 4.3. Post-Transcriptional Regulation by miRNAs and lncRNAs

MicroRNAs (miRNAs) and long non-coding RNAs (lncRNAs) are pivotal regulators of bone homeostasis, exerting their effects through targeted modulation of E3 ubiquitin ligases and deubiquitinating enzymes (DUBs). For example, miR-497-5p enhances osteoblast differentiation by suppressing Smurf2, which stabilizes the TGF-β/Smad pathway and promotes osteogenic activity [98]. Similarly, miR-136-5p facilitates osteogenesis by inhibiting Smurf1, thereby preserving key osteogenic signaling molecules such as Smad1/5 and Runx2 to maintain bone matrix synthesis [99]. In contrast, miR-708-5p delays osteoblast differentiation through its inhibition of SMAD-specific E3 ligase 2, serving as a regulatory brake to balance proliferation and differentiation processes [100].

The regulatory network extends further with miR-19b, which promotes osteogenic differentiation and fracture repair by targeting both WWP1 and Smurf2. This dual inhibition stabilizes KLF5 and activates the Wnt/β-catenin pathway, driving bone formation [101]. Conversely, the lncRNA MEG3 suppresses osteogenesis by downregulating miR-543, leading to indirect upregulation of SMURF1 and subsequent disruption of osteoblast activity [102]. Notably, miR-101, encapsulated in MSC-derived exosomes, accelerates osteogenic differentiation by inhibiting FBXW7. This action elevates HIF1α and *FOXP3* levels, further enhancing bone formation [37].

Together, these non-coding RNAs exemplify a dynamic equilibrium in bone homeostasis: miR-497-5p, miR-136-5p, miR-19b, and miR-101 act as pro-osteogenic drivers, while miR-708-5p and MEG3 provide counter-regulatory constraints. Their coordinated interplay underscores the complexity of bone remodeling and highlights potential therapeutic avenues for addressing conditions such as osteoporosis and impaired fracture healing.

### 4.4. Environmental Factors

Microgravity environments have been shown to promote deubiquitination effects. Under microgravity conditions, the protein expression of CKIP-1, a key factor inhibiting bone formation, increases while its ubiquitination degree decreases, thereby inhibiting osteogenic activity [103]. This finding highlights the potential impact of environmental factors on bone health and underscores the importance of considering such factors in bone homeostasis research.

## 5. Discussion and Perspectives

### 5.1. Complexity and Diversity of E3 Ligases and DUBs in Bone Homeostasis

E3 ubiquitin ligases and deubiquitinating enzymes (DUBs) exhibit complex and diverse roles in regulating bone homeostasis. Different enzymes may exert opposite effects through the same signaling pathway or influence the same physiological processes through distinct mechanisms. For example, while many E3 ligases inhibit osteogenic differentiation by ubiquitinating and degrading osteogenic-related factors, certain E3 ligases can also inhibit negative regulators of osteogenesis, thereby promoting the osteogenic process [7,18,28,38]. Similarly, DUBs such as USP7 and USP10 can either promote or inhibit osteogenic differentiation depending on their specific substrates and signaling contexts [9,10].

Smurf1 and Smurf2 are prime examples of this complexity. Smurf1 predominantly degrades osteogenic factors like Smad1/5, BMP type I receptors, and Runx2, thereby inhibiting osteoblast differentiation and bone formation [7,8,26]. However, under specific conditions, Smurf1 can also promote osteogenic differentiation by targeting negative regulators. For instance, Smurf1 can degrade MEKK2, which is independent of BMP signaling, thereby modulating osteoblast function [104]. Smurf2, on the other hand, inhibits osteoblast differentiation by degrading KLF5 and other osteogenic regulators [28,29]. This dual role underscores the importance of context-dependent regulation in bone homeostasis [11].

The TRIM family also exemplifies this diversity. TRIM38 and TRIM16 promote osteoblast differentiation by stabilizing key transcription factors and signaling molecules involved in bone formation [17,18]. Conversely, TRIM21 inhibits osteoblast differentiation by targeting Akt for degradation, thereby reducing the activity of downstream signaling pathways that promote osteogenesis [31]. This highlights the importance of understanding the specific roles of each TRIM family member in bone homeostasis.

HERC proteins enhance osteoblast differentiation by regulating C-RAF levels and ERK/p38 phosphorylation [32,33,34]. Similarly, RSP5 promotes osteogenic differentiation through Akt ubiquitination, thereby activating downstream signaling pathways involved in bone formation [35]. These findings underscore the importance of these E3 ligases in promoting osteogenesis and maintaining bone health. Conversely, ligases like FBXW7 and CHIP inhibit osteogenesis by targeting HIF1α and RUNX2 for degradation, respectively [36,37,38]. FBXW7 inhibits osteogenic differentiation and cartilage formation by catalyzing the ubiquitination and degradation of HIF1α [36,37]. CHIP induces ubiquitin-dependent degradation of RUNX2 during osteoblast lineage development, interfering with osteogenic differentiation [38]. These findings highlight the importance of these E3 ligases in maintaining the balance between osteoblast differentiation and proliferation.

Interestingly, TRIM16 can decrease CHIP expression and increase CHIP ubiquitination, thereby stabilizing RUNX2 expression and promoting osteoblast differentiation [18]. This interaction underscores the importance of TRIM16 in modulating RUNX2 stability and promoting osteogenesis. By modulating the expression and activity of these E3 ligases, TRIM16 plays a crucial role in maintaining the balance between osteoblast differentiation and proliferation [105].

Of particular interest is the nascent field of ubiquitination in osteocytes. Current evidence suggests mechanical loading alters vesicle secretion via *CUL3* [88], while SQSTM1-mediated autophagy may remodel osteocyte dendrites [89]. These discoveries bridge the gap between bone mechanosensation and ubiquitin-dependent signaling.

### 5.2. Future Research Directions

Future research should focus on several key areas to further elucidate the roles of E3 ubiquitin ligases and DUBs in bone homeostasis and to identify potential therapeutic targets for bone diseases.

Continued efforts are needed to identify new E3 ubiquitin ligases and deubiquitinating enzymes involved in bone homeostasis. This will help improve our understanding of the regulatory network and identify potential therapeutic targets. For example, recent studies have identified several novel E3 ligases and DUBs with potential roles in bone formation and resorption [6,51]. Further characterization of these enzymes will provide new insights into the mechanisms underlying bone homeostasis. Further research is required to explore the functions of these enzymes under overall physiological and pathological conditions. This includes investigating their specific roles in skeletal development, fracture healing, osteoporosis, and other bone diseases. For example, recent studies have shown that dysregulation of specific E3 ligases and DUBs is associated with osteoporosis and fracture non-union [5]. Understanding the mechanisms by which these enzymes contribute to these conditions will provide important theoretical bases for developing novel treatment strategies.

Given the complexity of the skeletal system, it is essential to consider the influence of multiple factors, such as genetic background, environmental factors, and lifestyle, to fully understand the regulatory mechanisms of bone homeostasis. For example, recent studies have shown that microgravity environments can promote deubiquitination effects, thereby inhibiting osteogenic activity [103]. This highlights the importance of considering environmental factors in bone homeostasis research. Based on existing research findings, novel therapeutic strategies against bone diseases should be developed. This includes the development of small molecules, antibodies, or gene therapies targeting specific E3 ligases or DUBs to modulate bone formation and resorption. For example, inhibitors or activators of specific E3 ligases or DUBs could be developed to modulate bone homeostasis, offering new therapeutic options for patients with bone diseases [5].

Future research should focus on developing small molecules or antibodies that specifically target Smurf1 and Smurf2 to modulate their activity in bone diseases. Additionally, gene therapy approaches targeting key regulatory enzymes like USP7 and USP10 could offer novel therapeutic strategies for osteoporosis [11].

## 6. Conclusions

This comprehensive review highlights the intricate roles of E3 ubiquitin ligases and DUBs in bone homeostasis. E3 ligases, such as Smurf1/2, TRIMs, and FBXW7, regulate osteoblast differentiation and osteoclastogenesis by targeting key signaling molecules (e.g., Smads, RUNX2, Akt) for degradation. Conversely, DUBs like USP2, USP34, and OTUB1 stabilize osteogenic factors (e.g., β-catenin, Smad1) to promote bone formation. The duality of these enzymes is evident: Smurf1/2 predominantly inhibit osteogenesis but may enhance it under specific contexts, while USP7 exhibits substrate-dependent pro- or anti-osteogenic effects. miRNAs (e.g., miR-20a-5p, miR-136-5p) and environmental factors (e.g., microgravity) further modulate these pathways. Clinically, dysregulation of ubiquitination machinery is linked to osteoporosis and impaired fracture healing, emphasizing their potential as therapeutic targets. Future directions include elucidating novel enzymes, exploring their roles in pathological conditions, and developing targeted therapies (e.g., inhibitors of Smurf1/2 or activators of USP34) to restore bone homeostasis. This field promises transformative insights into bone disease mechanisms and precision medicine applications.

## Figures and Tables

**Figure 1 biomolecules-15-00679-f001:**
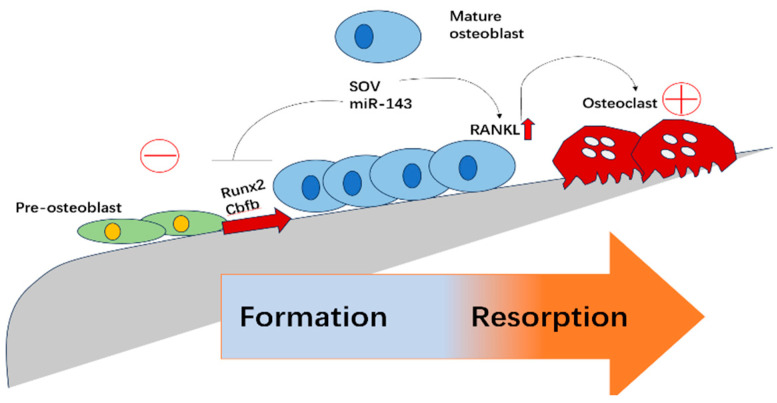
Schematic representation of osteoblast differentiation and its regulation of osteoclastogenesis. Pre-osteoblasts differentiate into mature osteoblasts through the activation of key transcription factors such as Runx2 and Cbfb. The process is modulated by regulatory molecules including SOV and miR-143. Mature osteoblasts secrete RANKL, which promotes osteoclast differentiation and bone resorption. This figure illustrates the dynamic balance between bone formation and resorption, highlighting potential regulatory checkpoints.

**Figure 2 biomolecules-15-00679-f002:**
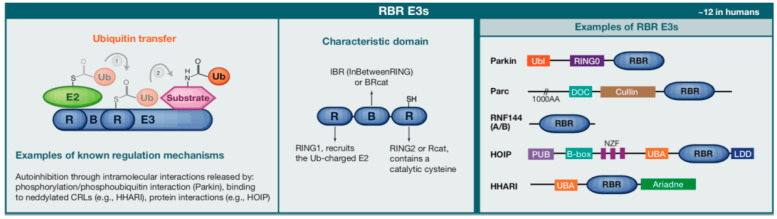
Structure, mechanism, and representative members of RBR E3 ubiquitin ligases. RBR (RING-between-RING) E3 ligases mediate ubiquitin transfer through a sequential mechanism involving E2 recruitment by RING1 and ubiquitin transfer via the catalytic cysteine in RING2. The IBR (In-Between RING) domain connects RING1 and RING2. Known regulatory mechanisms include autoinhibition relieved by phosphorylation (e.g., Parkin) or protein-protein interactions (e.g., HOIP). Examples of human RBR E3 ligases (~12 in total) include Parkin, Parc, RNF144A/B, HOIP, and HHARI, each containing unique auxiliary domains such as Ubl, DOC, PUB, and Ariadne, contributing to their specific regulatory functions.

**Figure 3 biomolecules-15-00679-f003:**
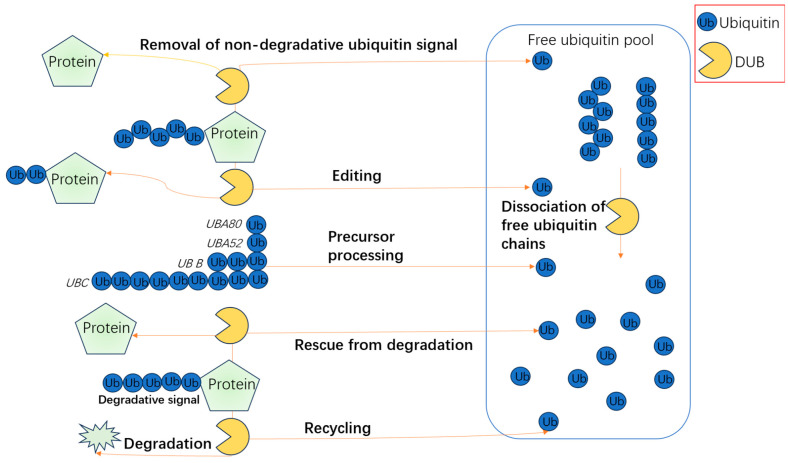
Multifunctional roles of deubiquitinating enzymes (DUBs) in ubiquitin homeostasis and protein regulation. DUBs mediate several key processes in the ubiquitin system. These include the removal of non-degradative ubiquitin signals to restore protein function, editing of ubiquitin chains to regulate signaling outcomes, processing of ubiquitin precursors (e.g., UBA52, UBA80, UBB, UBC) to generate free ubiquitin, and rescuing proteins from proteasomal degradation. In addition, DUBs dissociate free ubiquitin chains and recycle ubiquitin molecules into the free ubiquitin pool, ensuring a steady supply of functional ubiquitin for cellular processes.

**Figure 4 biomolecules-15-00679-f004:**
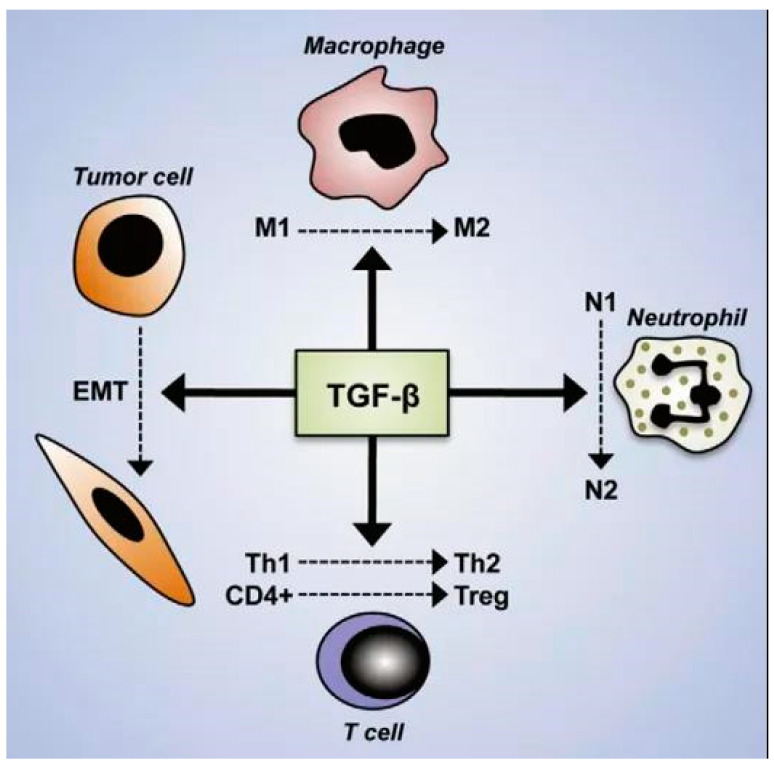
Transforming growth factor-beta (TGF-β) orchestrates the immunosuppressive tumor microenvironment by modulating immune cell polarization. TGF-β promotes epithelial–mesenchymal transition (EMT) in tumor cells and polarizes macrophages from pro-inflammatory M1 to immunosuppressive M2 phenotypes. It shifts neutrophils from tumor-suppressive N1 to tumor-promoting N2 subtypes. In T cells, TGF-β suppresses Th1 responses while promoting the differentiation of Th2 and regulatory T cells (Tregs), collectively contributing to immune evasion and tumor progression.

## Data Availability

Not applicable.

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
