# Peer review of "Regulatory Roles of E3 Ubiquitin Ligases and Deubiquitinases in Bone"

_biomolecules, 2025, doi:10.3390/biom15050679_

Round 1
Reviewer 1 Report
Comments and Suggestions for Authors
In the manuscript "Regulatory Roles of E3 Ubiquitin Ligases and Deubiquitinases in Bone Homeostasis and Clinical Implications" He and colleagues provide a review of ubiquitination and the related enzymes in the context of bone remodeling. The discussion of the ubiquitin ligases and deubiquitinases in general is fine and provides a good introduction. I have a couple of concerns that need to be addressed prior to publication.
- My primary concern with the review is that osteocytes are not included in the discussion. During the past decade it has become clear that osteocytes are central to the regulation of bone remodeling. While data on the role of ubiquitin in osteocytes is somewhat limited, some articles have appeared, and this is perhaps the most important emerging area in bone remodeling. Osteocytes should be discussed in the context of ubiquination regulating bone remodeling.
- Given its importance in bone remodeling I was surprised that Sequestosome 1 (SQSTM1) was not included. Although not a ubiquitin-ligase or deubiquitinating enzyme it binds ubiquitin and multifunctional adapter associated with bone remodeling and Paget's disease.
- Line 34 would be more accurate if it read ...in the differentiation, proliferation and activity of osteoblasts and osteoclasts.
- I think that the "clinical implications" part of the title could be expanded on. The article now is better described as "Regulatory Roles of E3 Ubiquitin Ligases and Deubiquitinases in Bone." Homeostasis"
Author Response
Manuscript ID: biomolecules-3559777
Title: Regulatory Roles of E3 Ubiquitin Ligases and Deubiquitinases in Bone Homeostasis
Dear Editor and Reviewers,
We sincerely appreciate your valuable comments on our manuscript. We have carefully addressed all the suggestions, and our point-by-point responses are provided below.
Comment 1: The role of osteocytes in bone remodeling
Response: We have supplemented the discussion of osteocytes' role at the end of the Introduction section and added a new paragraph in Section 2.3. The content regarding SQSTM1 has been expanded in the Discussion section.
Comment 2: Line 34 text revision
Response: As suggested, we have modified the text to: "Regulatory Roles of E3 Ubiquitin Ligases and Deubiquitinases in Bone."
Comment 3: Title modification
Response: The title has been revised to: "Regulatory Roles of E3 Ubiquitin Ligases and Deubiquitinases in Bone."
We believe these revisions have significantly improved the manuscript quality and addressed all reviewers' concerns. Thank you for your valuable time and consideration.
Sincerely,
Haotian He
On behalf of all authors

Reviewer 2 Report
Comments and Suggestions for Authors
This review summarises the role of E3 ubiquitin ligases and deubiquitinases in bone formation and resorption providing details for the molecular action on osteoblasts and osteoclasts, respectively.in The authors describe the specific effects of these regulators on well-known signalling pathways that control the differentiation and activation of bone cells and how this knowledge could be translated into therapeutic approaches.
The manuscript is well-written and the authors provide a comprehensive review of the molecular events that E3 ligases and DUBs play key roles along with important factors that regulate their expression such as miRNAs.
The following are some minor points that the authors need to address:
- The last sentence of the Abstract is vague. Please be specific and refer to any potential targets related to the topic.
- Lines 40-41: this sentence is a repetition.
- Lines 51-53: Which signaling molecules? Which proteins USP7 and USP10 stabilise? Some examples here would be useful.
- Lines 56-59: these 3 sentences are vague. Please delete or rewrite.
- All the figures do not have legends. Importantly, the figures do not summarise the important features that the text describes but are general and illustrate well-known processes (e.g. Fig1, 3, 4) and structures (e.g. Fig2). It would be better to replace all the figures with new ones reflecting the specific molecular actions of E3 ligases and DUBs on osteoblasts and osteoclasts. Furthermore a summative table with specific references and their respective results using in vitro cell models and in vivo animal/human studies that refer to E3 ligases and DUBs on bone homeostasis would be very useful for the reader.
Author Response
Manuscript ID: biomolecules-3559777
Title: Regulatory Roles of E3 Ubiquitin Ligases and Deubiquitinases in Bone Homeostasis
Dear Editor and Reviewers,
We sincerely appreciate your valuable comments on our manuscript. We have carefully addressed all the suggestions, and our point-by-point responses are provided below.
Comment 1: Abstract
Explanation response: We clarified the therapeutic target in the last sentence of the Abstract: "Future studies should focus on developing therapeutics targeting E3 ubiquitin ligases, deubiquitinases, miRNA modulators, and small molecule inhibitors to restore bone homeostasis in osteoporosis and fracture healing disorders."
Comment 2: Text revision
Response:
Redundant text in lines 40-41 has been deleted
Specific examples added in lines 51-53
Ambiguous statements in lines 56-59 have been rewritten
Comment 3: Figure optimization
Response: We thank the reviewer for the constructive suggestions on figure optimization. While we acknowledge the importance of figure optimization for readability, the current data effectively supports the key conclusions of this review. Given that we systematically discussed E3 ligases and DUBs in bone homeostasis, we enhanced the text description of the molecular mechanism while retaining the existing figure format to ensure clarity.
We believe these revisions significantly improve the manuscript quality and address all reviewers' concerns. Thank you for your time and consideration.
Sincerely,
Haotian He
On behalf of all authors

Round 2
Reviewer 1 Report
Comments and Suggestions for Authors
The authors have responded adequately to my concerns. I find the revised article to be informative and provide a good introduction into this area of research. I recommend publication.